# Self-Study-Based Informed Decision-Making Tool for Empowerment of Treatment Adherence Among Chronic Heart Failure Patients—A Pilot Study

**DOI:** 10.3390/healthcare13060685

**Published:** 2025-03-20

**Authors:** Lea Iten, Kevin Selby, Celine Glauser, Sara Schukraft, Roger Hullin

**Affiliations:** 1Cardiology, Cardiovascular Department, University Hospital, University of Lausanne, 1011 Lausanne, Switzerland; lea.iten@chuv.ch (L.I.); celine.glauser@chuv.ch (C.G.); sara.schukraft@chuv.ch (S.S.); 2Center for Primary Care and Public Health, University of Lausanne, 1011 Lausanne, Switzerland; kevin.selby@unisante.ch

**Keywords:** heart failure, medication adherence, self-study, informed decision-making

## Abstract

**Background:** Adherence to drug prescriptions is often suboptimal among heart failure (HF) patients. Informed decision-making may improve patients’ adherence to HF drug prescriptions. **Aims of the study:** We aimed to test whether a self-study-based informed decision-making tool could improve adherence to drug prescriptions among ambulatory HF patients. **Methods:** A tool and a statement-based questionnaire were developed to evaluate drug adherence willingness based on COMPAR-EU recommendations. The test group (*n* = 40) was exposed to the tool + questionnaire; controls (*n* = 40) answered the questionnaire only. Agreement with statements of the questionnaire was graded on a scale of 0 to 4 points, reflecting no to full agreement. **Results:** The median age of controls was younger (56 vs. 61 years; *p* = 0.04); test and control group patients did not differ across other parameters (always *p* > 0.05). Patients in both groups agreed that “HF is a life-long disease” (3.5 vs. 4; *p* = 0.19) and that “only life-long drug treatment provides benefit” (4 vs. 4; *p* = 0.22). More test group patients confirmed improved comprehension of HF disease (3 vs. 2; *p* = 0.03) and greater acceptance that “achievement of benefit asks for a combination of HF drugs” (4 vs. 3; *p* = 0.009) and “daily intake” (4 vs. 3; *p* = 0.004). In test group patients, questions remained, resulting in less agreement that “all aspects of my heart disease” are understood (1 vs. 3; *p* < 0.001). Willingness to adhere to HF-drug treatment was not different between the groups (3 vs. 3.5; *p* = 0.28). **Conclusions:** The self-study-based informed decision-making tool improved the comprehension of HF and the need for HF treatment, but did not improve willingness to adhere since questions remained unanswered.

## 1. Introduction

Heart failure (HF) is a major health problem due to its high prevalence and the progressive nature of the disease [1]. It is the medical cause for 1–2% of all admissions at emergency departments worldwide, with nearly 80% of these acute HF patients requiring hospitalization for HF treatment (HHF) [2]. After discharge from HHF, about 25% of these patients are readmitted within 30 days, and up to 46% after 6 months [3].

HHF is a distressing and uncomfortable experience and is associated with physical and psychic trauma. After discharge, patients often experience a reduced quality of life, depression, and emotional distress due to their intuitive understanding that HF is associated with an increased risk of morbidity and premature death [4,5].

Despite this incisive experience, adherence to HF drug prescriptions remains suboptimal whether HF patients participate in clinical trials [6], national registries [7], or local cohorts [8]. This suggests that non-adherence occurs independent of the clinical setting, underlining the need to reach the individual patient.

In the past, multiple studies have tested varied combinations of education modules and follow-up strategies for the improvement of HF patients’ adherence. Education modules were either self-study-based or guided by healthcare professionals; the mode of follow-up has varied considerably, ranging from motivational messages to electronic adherence reminders, telephone contact and nurse-based home monitoring of HF signs [9,10]. The results have been heterogeneous but positive on the whole, as suggested by a recent meta-analysis [11].

However, it remains unclear why measures aimed at increasing adherence fail for individual patients. Informed decision-making tools contain elements that impart knowledge, help better define one’s intentions, and facilitate the development of realistic expectations based on the clarification of one’s short- and long-term goals. These elements can strengthen a patient’s control over the decision-making process since positive and negative effects, as well as the rightness of decision-making, become more transparent [12].

The primary goal of this study was therefore to test the applicability of a self-study-based information tool. In addition, the questionnaire also intended to analyze the extent to which non-informatory elements impact the readiness for decision-making.

## 2. Materials and Methods

### 2.1. Objective

The development and testing of a self-study-based informed decision-making tool intended to empower decisions to adhere to HF drug prescriptions.

### 2.2. Study Design

A mono-center, single-blinded clinical study comparing a test group (tool exposure + questionnaire) with control group patients (questionnaire only). The groups were tested sequentially to ensure the single blinding of study participants.

### 2.3. Outcome Measure

The effect of the self-study-based informed decision-making tool was measured using a statement-based questionnaire inquiring about factors and facilitators known to empower decision-making.

### 2.4. Recruitment Site

Advanced quality of care center at a Swiss University hospital [12].

### 2.5. Study Population

The study participants were recruited from consecutive ambulatory chronic HF patients scheduled for a routine HF follow-up. Patients were screened ahead of their planned visit; eligible patients were invited for study participation.

### 2.6. Inclusion Criteria

Patients were required to have clinical signs and symptoms of HF and echocardiographic characteristics typical of HF [13]. Patients were eligible when in stable chronic HF and ≥18 years of age; language ability had to be adequate. Moreover, study participants had to be undergoing routine follow-up at the outpatient HF clinic, with at least one preceding ambulatory HF visit.

### 2.7. Exclusion Criteria

Acute chest pain, signs and symptoms of worsening HF necessitating direct hospitalization, acute metabolic disorder, severe concomitant illness reducing survival expectancy to <12 months as judged by the study investigators, planned surgical intervention, cognitive impairment. Patients with heart transplantation or on mechanical circulatory support systems were not included.

### 2.8. Tool

The self-study-informed decision tool and the statements of the semi-structured interview were developed based on the COMPAR-EU recommendations for self-management in heart failure. Development was realized via a multidisciplinary collaboration of cardiologists (LI, RH), an internist with expertise in shared decision-making (KS), and an HFNP (CD).

The self-study informed decision-making tool consisted of three parts: the first part provided general information about HF; the second part reported foundational HF drugs and drugs known to improve symptoms. For each drug, essential benefits and important side effects were explained in common parlance, with one key message and one typical side effect per item. The third part stated that only the combination of all drugs enables benefit.

Acknowledging limitations related to the inclusion of non-native French speakers or old study participants, visual support was included into the decision-making tool (Appendix A).

The questionnaire aimed to evaluate factors necessary to implement sustained decisions as identified by the recommendations of the COMPAR-EU project [14,15]; statements 1 to 3 investigated decision tool factors, in particular, the understanding of the information that “HF is a whole life duration disease” and that “HF treatment with foundational drugs has beneficial short- and long-term effects”; statement 4 investigated individual health professional factors, asking whether the “study participant’s appraisal of the benefit of life-long medical adherence is compatible with the individual’s vision of his life”; statements 5 and 6 explored interaction and organizational factors such as whether comprehension is sufficient to take a decision, or whether additional information is afforded; statement 7 tested study social factors, in particular, participant’s readiness to take a decision. The wording of statements 5 and 6 differed between the test and the control groups in order to acknowledge exposure/non-exposure to the self-study-informed decision-making tool (Appendix A). In the control and test questionnaire, all statements but statement 5 were similar. In the control group, statement 5 was “I understand the different aspects of my heart disease”, whereas test group statement 5 was “Following the presentation, I have new questions that remain unanswered”. Both statements evaluated the need for further information, taking into consideration that only the test group participants were exposed to the information tool. Using the Likert scale, each statement was graded for agreement (grades of agreement: 0 = not at all, 1 = rather no, 2 = rather yes, 3 = yes, 4 = absolutely). The total score is the sum of the grades for the 7 questions. Patients were encouraged to provide written feedback and comments (Appendix A).

### 2.9. Study Execution

The study was conducted after approval by the local ethics committee. Between 1 April 2022 to 31 January 2023, ambulatory stable HF patients were screened ahead of their scheduled routine visit; eligible patients were invited by phone for study participation about two weeks before the planned routine visit (Figure 1).

For the test group participants, the heart failure nurse practitioner (HFNP) assisted the self-study of the informed decision-making tool and execution of the statement-based questionnaire in order to support text understanding. The controls were exposed to the questionnaire only (central illustration). Answers to questions regarding medical issues were provided by the HFNP only after completion of the questionnaire (see respective recordings in Appendix A).

### 2.10. Statistical Analysis

Sample size was chosen in accordance with the data in the published literature, which tested interactive educational tools in a single test group setting including 42 participants [16], and in a setting of a test and a control group with each group including each 38 HF participants [17]. Continuous variables were expressed as median interquartile range; categorical variables were expressed as frequencies and percentages. Normality was assessed by the visual inspection of histograms and the computation of Q-Q plots. Continuous variables have been analyzed using the Student *t*-test or the Wilcoxon rank-sum test per distribution. Categorical variables were compared using chi-square or Fisher exact test as appropriate. For the comparison of statement 5, the score of statement 5 in the test group was inverted, as foreseen in the study plan. All statistical analyses were performed using dedicated software (Stata 14.2, College Station, TX, USA) at a 2-tailed significance level of alpha = 0.05.

## 3. Results

Figure 1 shows that 82 of 116 screened candidates agreed to participate. Table 1 shows baseline characteristics of the study participants in the test and the control group. Male sex was predominant in both groups (intervention group vs. control group: 75 vs. 80%, *p* = 0.79); the median age was higher in the test group (61 vs. 56 years; *p* = 0.04). There was no difference between groups regarding the prevalence of HF of ischemic or non-ischemic origin (test group vs. control group: 25 vs. 32%, *p* = 0.62; 55 vs. 45%, *p* = 0.50; respectively), the HF severity graded by New York Heart Association (NYHA) functional class (*p* > 0.48), or the number of participants in the three HF subgroups (*p* > 0.58).

The median time after diagnosis with HF was not different between groups (4 vs. 5.5 years; *p* = 0.61). Figure 2a shows the distribution of the time living with HF diagnosis for all study participants; within each group, the portions of study participants living with the diagnosis for <5 years versus ≥5 years were not different (Figure 2b).

The prevalence of cardiovascular risk factors and the median total number of comorbidities per patient group were not different between groups. Furthermore, there was no difference between groups regarding the number of patients with more severe valvopathy, implanted devices, or previous cardiac bypass surgery; 3 patients in the test group were cardiac arrest survivors (Table 1).

Table 2 shows that the prescription of the foundational pharmacological classes or diuretic treatment did not differ between groups. Likewise, the percentages of patients on the recommended maximal drug doses were not different for either pharmacological class (always *p* > 0.17). The mean number of HF drug classes prescribed per patient was equal in both groups (3 vs. 3; *p* = 0.81).

There was no difference in the total score either between groups (22 vs. 22 points; *p* = 0.65) (Figure 2, Appendix A) or for the distribution of the individual patient’s total score within the groups (Appendix A).

Table 3 shows there is no difference when comparing scores for statement 1 (addressing the comprehension that HF is a chronic life-long disease) as well as for statement 2 (asking about the understanding that the foundational drug treatment of HF disease is aimed at patients’ well-being). More study participants in the test group agreed that only the combination of HF drugs delays the progression of HF disease (statement 3: 4 vs. 3 points; *p* = 0.009). More patients in the test group agreed that the daily intake of the HF drugs is consistent with their vision of life (statement 4: 4 vs. 3 points, *p* = 0.004). However, study participants in the test group had unanswered questions and aspects that remained unclear (Appendix A), and agreed less to the statement “I understand the different aspects of my disease” when compared to controls (statement 5: 1 vs. 3 points, *p* < 0.001). However, test group patients confirmed an improved understanding of HF disease (statement 6: 3 vs. 2 points, p= 0.03). Willingness to take an informed decision in favor of adherence to HF drug prescription was not different between groups (statement 7: 3 vs. 3.5 points, *p* = 0.28).

When the 80 individual scores of agreement with statement 7 were further investigated, patients with scores of 0 to 3 points had a lower median number of HF drug classes when compared to patients with 4 points (statement 7: 3 vs. 4; *p* = 0.03). However, if study participants were placed on a treatment with four pharmacological classes (in the test group *n* = 15; in the control group *n* = 17), the number of patients scoring 4 points (=feeling fully able to make informed decision on adherence management) was not different (test vs. control group: 10/15 vs. 10/17 study participants; *p* = 0.73).

## 4. Discussion

After exposure to the self-study-informed decision tool, more HF patients agreed that a combination of different HF drugs slows the progression of HF disease, and that daily medication is compatible with their vision of life. However, questions remained unanswered after exposure to the tool due to the self-study design. This limitation can explain why willingness to take a decision regarding HF drug adherence was not different between groups, despite the significant positivity of aspects favoring willingness to adhere.

HF patients participating in multi-disciplinary management programs not only benefit from evidenced-based therapy, but also from education and follow-up [13]. Education and the type of follow-up vary largely as a function of regional or national reality; however, in spite of variability, they reduce all-cause mortality [11,18]. The role of self-management strategies is less clear, although the reduction in risk for HHF or all-cause mortality in younger HF patients suggests usefulness [11,19]. In contrast, the application of education tools alone did not reduce outcomes in HF patients [18]. The European Guidelines for the care of the HF patients nonetheless underline the importance of patient education because of their potential to decrease misunderstandings and misconceptions of the severity of HF disease due to a lack of knowledge [13]. In addition, it is believed that patient education provides a chance for self-care empowerment [20,21].

So far, education-based informed decision-making is not proposed by the guidelines despite the recognition that informed decision-making facilitates willful decisions because of the integration of individual and social values, along with elements helping the patient to deliberate over benefits and risks [22]. These elements are important since their insufficient appreciation may result in decisional conflict [23]. The integration of such personal elements into informed decision-making patient education therefore has the potential to improve self-management based on willful decision [24].

The self-study-informed decision-making tool of this study features decisional, professional, interaction, and organizational factors [14,22,23], starting with decision tool factors in the form of information regarding the four foundational HF drug classes, diuretics and the I_f_ channel blocker [13]. To reinforce this information, the tool cites evidence from the literature proving that clinical trials with thousands of patients increase survival and reduce the risk of HHF [6,7,8,13]. In addition, principal side-effects are presented to identify decision-associated risks, thus facilitating the appreciation of personal advantages and disadvantages [14,23]. While this information provides decision tool factors, the use of images of respective pills and pill boxes aimed to strengthen individual health professional factors considering that the majority of study participants were already treated with HF drugs. This visualization aimed at facilitating the realization that the presented HF drug treatment corresponds to drugs already prescribed. Last but not least, the questionnaire investigated whether the intake of foundational drugs corresponds with patients’ values and personal resources, addressing the compatibility of HF drug treatment adherence with patients’ personal vision of life.

These features distinguish the present decision-making tool from other education tools presenting HF-related information based on animations and computer-based voice-overs [25], icon-guided interactions [26], or web-based HF education platforms such as www.heartfailurematters.org. The effects of these tools have not been evaluated prospectively; however, increases in HF knowledge and the improvement of self-care behavior are suggested in a study exposing HF patients in ambulatory follow-up to an interactive educational board game [17]. The present self-study-informed decision-making tool is therefore unique, and provides the first insights into self-study-based informed-decision making based on COMPAR-EU recommendations and its potential role in the care of HF patients.

After self-study, the study participants in the test group gave higher scores for the statements that “The combination of the different treatments helps to slow disease progression” and that “Daily medication is in agreement with my vision of the disease”. Of note, confirmation that HF drug treatment is compatible with the patients’ vision of life was provided after appreciation of the risks and benefits of HF treatment, suggesting coherence between decisions and individual health professional factors [14].

However, a willful decision regarding treatment adherence was not empowered in the test group, which is paradoxal in view of the coherence of decision and health professional factors. In fact, the test group patients had “new questions that remain unanswered”, suggesting that new unanswered questions affected the rightness of the time to decide. This is in accordance with personal notes asking for “more details about progression of the disease, medical treatment, or adequate lifestyle with HF” in test group patients. Moreover, the exploration of the 37 of the 80 study participants agreeing in full with statement 7, “capable of making a decision”, shows that all were under treatment with all four of the foundational drugs. Since the portions of study participants under treatment with the four foundational drugs were not different between the test group and controls, we can exclude that disproportion of the number of HF patients on full HF treatment was the reason for this observation. Thus, test group study participants not under treatment with the four foundational drugs and not given the chance to receive answers to questions because of the self-study design may have felt unsure; even more so since they had the information that all four drugs are considered necessary for successful HF treatment. These observations indicate that patients with remaining questions should be given the chance of a face-to-face with a health professional to receive answers to their questions and thus arrive at a willful decision. This conclusion is in accordance with the observation that nurse-based education at discharge from HF-related hospitalization increases self-care and adherence to treatment, and decreases the risk of HHF [27,28].

### Limitations

The statistical analysis remained descriptive due to the small sample size, which does not permit the deduction of relationships. Nonetheless, the results indicate the applicability of the tool while suggesting that a face-to-face exchange with a healthcare professional should be available if questions remain after self-study, or if patients are of old age. With this addendum, and ahead of broad application, the tool should be tested in a larger study population in order to quantify the impact on drug adherence.

Another limitation of this pilot study is the fact that the tool focuses on HF drugs without considering complementary recommendations for HF treatment, such as salt or water restriction. Likewise, aspects addressing patients’ personal needs, organizational factors accounting for social limitations, or self-care [29] were not accounted for. The reason for this narrow focus on HF drugs was the multidisciplinary approach to the care of HF patients at our institution, with hospital- or home-based nurses taking care of many of these aspects. Therefore, healthcare professionals might consider the integration of these aspects in a local version of the informed-decision tool if the clinical environment at their institution is different.

## 5. Conclusion

The results of this study show that a self-study-based informed decision-making tool increases the understanding of and agreement with essential elements of HF treatment. In the test group participants, self-study was associated with the presence of remaining questions, suggesting that a face-to-face encounter with a health profession would complement the self-study tool. The addendum of a face-to face encounter may question the importance of the self-study-based information tool, but a self-study-based information tool not only imparts knowledge that by itself can decrease intentional non-adherence [30], as self-study also helps to define one’s own intentions, and facilitate the development of realistic expectations based on the clarification of one’s own short- and long-term goals. Therefore, a face-to-face encounter will answer knowledge-related questions, but likewise provides the option to consider the multimodality of the decision-making process to improve timing in decision-taking (Figure 3).

## Figures and Tables

**Figure 1 healthcare-13-00685-f001:**
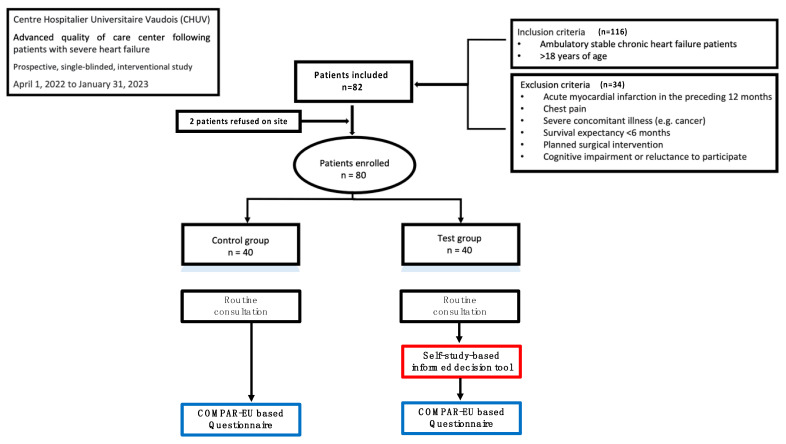
Flow diagram of the mono-center, single-blinded prospective study. The participants were recruited from consecutive chronic severe HF patients scheduled for routine ambulatory follow-up. The patients were screened ahead of their planned visit and were invited for study participation if eligible. Groups were tested sequentially to avoid crosstalk in the waiting room. Black = routine control; red = intervention; blue = evaluation.

**Figure 2 healthcare-13-00685-f002:**
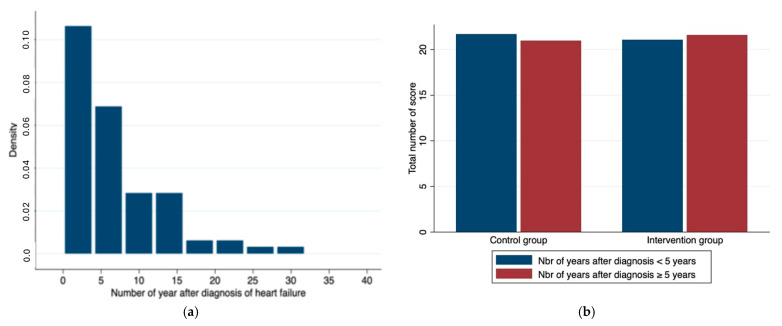
(**a**) Distribution of the number of years after HF diagnosis in all study participants. (**b**) Mean total score per group of patients in each group analyzed as a function of the number (Nbr) of years with HF (heart failure) diagnosis (<5 years versus ≥5 years). The mean duration of HF was not different between the intervention vs. the control group.

**Figure 3 healthcare-13-00685-f003:**
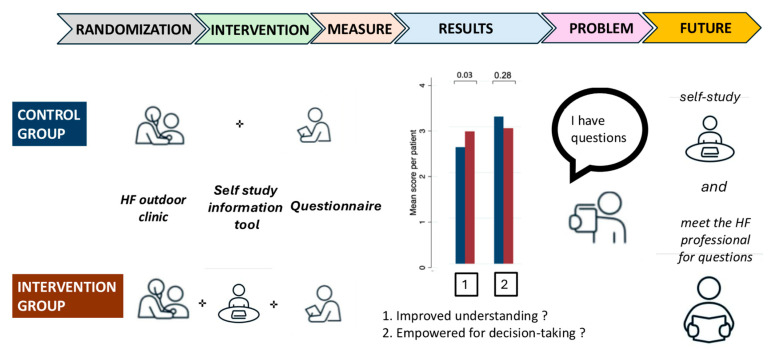
Central illustration.

**Table 1 healthcare-13-00685-t001:** Demographics and clinical characteristics of the study population.

	Control Group (*n* = 40)	Test Group (*n* = 40)	*p*-Value
Male, *n* (%)	32 (80)	30 (75)	0.79
Age, y	56 (47–62)	61 (53–68)	0.04
Natives French study participants (*n*; %)	28 (70)	31 (78)	0.61
ETIOLOGY			
Ischemic only (*n*; %)	13 (32)	10(25)	0.62
Mixed origin (*n*; %)	9 (22)	8 (20)	1.00
Non-ischemic (*n*; %)	18 (45)	22 (55)	0.50
Heart failure subgroup			
<40% (*n*; %)	30 (75)	33 (83)	0.59
41–49% (*n*; %)	6 (15)	5 (13)	1.00
>50% (*n*; %)	4 (10)	2 (5)	0.68
Clinical severity			
NYHA I (*n*; %)	16 (40)	17 (43)	1
NYHA II (*n*; %)	13 (33)	17 (43)	0.49
NYHA III (*n*; %)	9 (23)	6 (15)	0.57
NYHA IV (*n*; %)	1 (3)	0 (0)	1
Number of years after diagnosis (y)	4 (2.5–9.5)	5.5 (2–8.5)	0.61
Cardiovascular risk factors			
Smoking (*n*; %)	7 (18)	10 (25)	0.59
Arterial hypertension (*n*; %)	32 (80)	27 (68)	0.31
Dyslipidemia (*n*; %)	31(78)	26 (65)	0.32
Positive family history (*n*; %)	0 (0)	2 (8)	0.24
Number of Cardiovascular RF (*n*)	2 (1–3)	2 (1–3)	0.77
Others			
CABG (*n*; %)	4 (10)	7 (18)	0.52
Valvopathy (*n*; %)	31 (76)	23 (58)	0.09
Defibrillator (*n*; %)	19 (48)	28 (45)	1
Resynchronization therapy (*n*; %)	13 (33)	8 (20)	0.31
Previous cardiorespiratory arrest (*n*; %)	0 (0)	3 (8)	0.24

Legend: Data are presented as numbers (*n*), percentage (%). Years are presented as the mean with 25–75% a percentage interval. CABG: Coronary artery bypass graft. Mixed: ischemic and other cause. Non-ischemic: hypertension and/or valve disease and/or arrhythmia and/or genetic cardiomyopathy and/or congenital heart disease and/or drug-induced. NYHA: New York Heart Association functional classification. valvopathy: valve insufficiency with grade ≥ 2/4 or prosthetic valve or Mitra clip.

**Table 2 healthcare-13-00685-t002:** Drug treatments of the study population.

	Control Group (*n* = 40)	Test Group (*n* = 40)	*p*-Value
Angiotensin converting enzyme inhibition	9 (22)	12 (30)	0.61
TD <25% (*n*; %)	0 (0)	0 (0)	
TD 25–49% (*n*; %)	2 (5)	2 (5)	1
TD 50–74% (*n*; %)	3 (8)	4 (10)	1
TD 75–100% (*n*; %)	4 (10)	6 (15)	0.74
Angiotensin II receptor blockade	12 (30)	6 (15)	0.18
TD <25% (*n*; %)	3 (8)	2 (5)	1
TD 25–49% (*n*; %)	1 (3)	2 (5)	1
TD 50–74% (*n*; %)	5 (13)	1 (3)	0.20
TD 75–100% (*n*; %)	3 (8)	1 (3)	0.62
Angiotensin receptor Neprilysin inhibition	17 (43)	18 (45)	1
TD <25% (*n*; %)	0 (0)	0 (0)	-
TD 25–49% (*n*; %)	1 (3)	2 (5)	1
TD 50–74% (*n*; %)	8 (20)	9 (23)	1
TD 75–100% (*n*; %)	8 (20)	7 (16)	1
ß-blocker treatment	32 (80)	32 (80)	1
TD <25% (*n*; %)	0 (0)	0 (0)	
TD 25–49% (*n*; %)	12 (30)	17 (43)	0.35
TD 50–74% (*n*; %)	7 (18)	7 (18)	1
TD 75–100% (*n*; %)	13 (33)	7 (18)	0.19
Mineralocorticoid receptor antagonists	25 (63)	28 (70)	0.6
TD <25% (*n*; %)	0 (0)	0 (0)	
TD 25–49% (*n*; %)	6 (15)	9 (23)	0.57
TD 50–74% (*n*; %)	9 (23)	14 (35)	0.32
TD 75–100% (*n*; %)	10 (25)	5 (13)	0.25
SGLT2 inhibition	24 (60)	20 (50)	0.5
TD <50% (*n*; %)	0 (0)	0 (0)	
TD 50–74% (*n*; %)	1 (3)	1 (3)	1
TD 75–100% (*n*; %)	23 (58)	19 (48)	0.50
Loop diuretic treatment (*n*; %)	13 (33)	17 (43)	0.49
Pharmacological classes per patient (*n*; %)	3 (2–4)	3 (2–4)	0.81

Legend: TD = target dose recommended by the ESC guidelines [13]. SGLT2 = sodium glucose transporter 2.

**Table 3 healthcare-13-00685-t003:** Scores of questionnaire statements.

	Control Group (*n* = 40)	Test Group (*n* = 40)	*p*-Value
Median sum of the points	22 (19–24)	22 (19–24)	0.65
Statement 1: HF is a chronic life-long disease	3.5 (3–4)	4 (3–4)	0.19
Statement 2: Foundational drug treatment of HF disease intends patient’s well-being	4 (3–4)	4 (4–4)	0.22
Statement 3: The combination of HF drugs delays progression of HF disease	3 (3–4)	4 (3–4)	0.009
Statement 4: Daily intake of HF drugs is consistent with the vision of HF	3 (3–4)	4 (3–4)	0.004
Statement 5: Controls: understanding of different aspects of my heart disease; test group: all questions are answered *	3 (2.5–4)	1 (0–1.5)	<0.001
Statement 6: Improved understanding the different aspects of disease	2 (1–4)	3 (2–4)	0.03
Statement 7: Empowerment to take an informed decision for adherence self-management	3.5 (3–4)	3 (2–4)	0.28

Using the Likert scale, each statement has been graded for agreement (grades of agreement: 0 = not at all, 1 = rather no, 2 = rather yes, 3 = yes, 4 = absolutely). Values provided represent the mean; the numbers in parenthesis represent the 25–75% percentile. * the original statement required augmentation with new questions after tool exposure when the controls were asked about their understanding of aspects of their heart disease; for comparison, the scoring in the test group was inversed.

## Data Availability

The original contributions presented in this study are included in the article/Appendix A. Further inquiries can be directed to the corresponding author.

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
