# Peer review of "Self-Study-Based Informed Decision-Making Tool for Empowerment of Treatment Adherence Among Chronic Heart Failure Patients—A Pilot Study"

_healthcare, 2025, doi:10.3390/healthcare13060685_

Round 1
Reviewer 1 Report
Comments and Suggestions for Authors
The introduction gives a decent overview of the problem, but it doesn’t explain clearly enough what makes this new tool different from other education strategies.
Why should anyone care about this particular approach? Highlighting what’s unique about it would make the introduction stronger.
The methods section is detailed, but it’s missing a few important things. For example, the authors say the sample size was based on previous studies but don’t explain why that number of participants is enough for reliable results. A proper justification, like a power calculation, would make the study better.
Older participants might find it harder to use self-study tools, which could have influenced the results. Also, the way one of the questionnaire statements was analised differently for the test group is a bit confusing and should be explained better.
The conclusion is short and to the point but could be more practical. It would be helpful to include advice on how this tool could be used in everyday healthcare.
Comments on the Quality of English Language
The English in the manuscript is functional but could use some polishing for clarity and fluency.
Author Response
The introduction gives a decent overview of the problem, but it doesn’t explain clearly enough what makes this new tool different from other education strategies.
Why should anyone care about this particular approach? Highlighting what’s unique about it would make the introduction stronger.
Response: We thank the reviewer for the comment and the strong question and rewrote the last two paragraphs of the introduction (lines 48-56):
However, it remains unclear why measures aiming to increase adherence fail in the individual patient. Informed decision-making tools contain elements which impart knowledge, help better define own's intentions, and facilitate development of realistic expectations based on clarification of own's short- and long-term goals. These elements can strengthen patient's control of the decision-making process since positive and negative effects as well as rightness of decision-making become more obvious [12].
The primary goal of this study was therefore to test applicability of the self-study-based information tool. In addition, the questionnaire also intended to analyze to what extent the non-informatory elements impact on the readiness for decision-taking.
Older participants might find it harder to use self-study tools, which could have influenced the results. Also, the way one of the questionnaire statements was analised differently for the test group is a bit confusing and should be explained better.
Response: We thank the reviewer for this comment and agree that age may affect the results. Test group patients were older, and more test group patients confirmed improved comprehension of heart failure disease and treatment (question 3,4,6) suggesting that at least in the context of this study age was not a barrier. The sample size being small, statistical analysis remained descriptive.
Furthermore, we changed the explanation to questions 5 in the test and the control group in the lines 114-119 to "In the control and test questionnaire, all statements but statement 5 were similar. In the control group, statement 5 was "I understand the different aspects of my heart disease" whereas test group statement 5 was "Following the presentation, I have new questions that remain unanswered". Both statements evaluated the need for further information taking into consideration that only the test group participants were exposed to the information tool."
The conclusion is short and to the point but could be more practical. It would be helpful to include advice on how this tool could be used in everyday healthcare.
Response: We agree with the reviewer and added to the conclusions (lines 299-311):
The results of this study show that a self-study-based informed decision-making tool increases understanding and agreement with essential elements of HF treatment. In the test group participants, self-study was associated with remaining questions suggesting that a face-to-face encounter with a health-profession should complement the self-study tool. The addendum of a face-to face encounter may question the importance of the self-study-based information tool. However, a self-study-based information tool not only imparts knowledge which by itself can decrease intentional non-adherence [29]. Self-study information tools based on the COMPAR-EU also help to define own's intentions and facilitate development of realistic expectations based on clarification of own's short- and long-term goals. Therefore, a face-to-face encounter will answer knowledge-related questions but likewise provides the option to consider the non-information related elements which are important to achieve rightness of the time for decision-taking (central illustration in the new version of the manuscript).
Reviewer 2 Report
Comments and Suggestions for Authors
The article has significant potential to contribute to the field of HF management by offering an innovative self-study tool for patient education. However, to maximize its impact and clinical applicability, the authors should consider enhancing data presentation, emphasizing practical applications, addressing the limitations of the self-study model, and directly measuring adherence outcomes in future studies. By adopting these recommendations, the article can provide a more comprehensive and actionable framework for improving HF care.
Author Response
The article has significant potential to contribute to the field of HF management by offering an innovative self-study tool for patient education.
Response: We thank the reviewer for this kind comment.
However, to maximize its impact and clinical applicability, the authors should consider enhancing data presentation, emphasizing practical applications, addressing the limitations of the self-study model, and directly measuring adherence outcomes in future studies.
Response: We thank the reviewer for this insightful comment and created this comprehensive data presentation (see central illustration in the manuscript)
By adopting these recommendations, the article can provide a more comprehensive and actionable framework for improving HF care.
Response: Thank You for this comment of appreciation.
Reviewer 3 Report
Comments and Suggestions for Authors
One of the primary weaknesses of this study lies in its limited impact on medication adherence, as the self-study tool improved comprehension but failed to translate into a significant behavioral change.
Additionally, the absence of a long-term follow-up restricts the understanding of whether the intervention could sustain adherence or improve clinical outcomes over time.
The small sample size, with only 40 participants in each group, further limits the statistical power and generalizability of the findings.
Furthermore, the study design’s reliance on self-study, without immediate access to healthcare professionals for addressing unresolved questions, may have hindered participants’ confidence in making informed decisions. The slight age difference between the test and control groups (median age: 61 vs. 56 years, p = 0.04) could also act as a confounding variable, potentially affecting outcomes.
Lastly, the narrow focus on HF medication adherence, without considering other critical aspects of disease management such as lifestyle changes or symptom monitoring, restricts the tool's holistic utility in heart failure care.
Author Response
One of the primary weaknesses of this study lies in its limited impact on medication adherence, as the self-study tool improved comprehension but failed to translate into a significant behavioral change.
Additionally, the absence of a long-term follow-up restricts the understanding of whether the intervention could sustain adherence or improve clinical outcomes over time.
Response: We thank the reviewer for this comment. The intention of this study was to investigate whether the readiness to adhere to heart failure treatment is improved after exposure to self-study-based information tool without implication of a HF professional. The present study was intended as a pilot-study testing applicability of the self-study-based information tool.
The results of the study confirm applicability of the self-study-based information tool which increased significantly comprehension of the importance of heart failure drug treatment (question 2-4). Furthermore, in the test group, acceptance that "daily intake of HF drugs is consistent with the individual's vision of life" (question 5) was significantly increased when compared the control group. However, readiness to take the decision for adherence failed to improve (question 7) because of remaining questions (question 6). This is an important result suggesting that remaining questions afford answer to achieve rightness of the time for decision-taking.
Questions can knowledge-related or may relate to questions addressing own's intentions and or expectations from clarification of own's short- and long-term goals on the basis of the self-study-based information tool. Therefore, a face-to-face encounter will be valuable complementation offering the opportunity to consider the multimodality of decision-making process to achieve rightness of the time for decision-taking.
Altogether, the pilot study provides the basis for a clinical study comparing adherence outcomes in study participants exposed to the self-study-based information tool with or without subsequent face-to-face encounter with the HF professional.
The small sample size, with only 40 participants in each group, further limits the statistical power and generalizability of the findings.
Response: We agree with the reviewer. However, we would like to point out that this study tested the self-study-based tool with respect to its applicability. The result of the study validates applicability (see answer questions 2-4). In addition, we identified by analysis of different dimensions of the decision-making process (see COMPAR-EU Krah et al.) why the self-study information tool alone failed to empower decision-taking. We agree that this pilot study cannot be taken and copy-pasted into routine follow-up of HF patients. However, the results serve as starting for a future clinical study aiming to provide broadly applicable results.
Furthermore, the study design’s reliance on self-study, without immediate access to healthcare professionals for addressing unresolved questions, may have hindered participants’ confidence in making informed decisions. The slight age difference between the test and control groups (median age: 61 vs. 56 years, p = 0.04) could also act as a confounding variable, potentially affecting outcomes.
Response: We agree with the reviewer for his insightful comment. Indeed, we cannot exclude that the slight age difference may have affected the study results. Because of the small study size, in-depth statistical analysis such as a multivariable analysis does not seem adequate.
Lastly, the narrow focus on HF medication adherence, without considering other critical aspects of disease management such as lifestyle changes or symptom monitoring, restricts the tool's holistic utility in heart failure care.
Response: We thank the reviewer for this comment. In our experience, other critical aspects of HF treatment (salt-intake, restriction of liquid consumption, physical exercise, and others) usually find broader acceptance while need of HF drugs and their potential side effects are routinely put into question resulting in intentional non-adherence in particular in HF patients without sufficient knowledge.
Reviewer 4 Report
Comments and Suggestions for Authors
In this manuscript (healthcare- 3379199), the authors studied the efficiency of self-study based informed decision-making on the inclination to follow heart failure drug prescription. The authors found that the self-study informed decision-making tool progressed understanding of HF and the requirement of HF treatment. However, it did not improve compliance to remain since questions remained unanswered.
Major concerns
- Please explain the novelty of this study.
- In the study have authors noticed any other metabolic disorders?
- The authors must cite more relevant references.
- In line 76 and 82, start sentences with capital letters.
- In figure 2 legends, format and styling should be same.
- In line 225, correct the spacing of usefulness.
- In table 1, correct the spelling of mixed origin.
Language can be improved.
Author Response
In this manuscript (healthcare- 3379199), the authors studied the efficiency of self-study based informed decision-making on the inclination to follow heart failure drug prescription. The authors found that the self-study informed decision-making tool progressed understanding of HF and the requirement of HF treatment. However, it did not improve compliance to remain since questions remained unanswered.
Response: We agree that this message is central for the manuscript. In the manuscript we state in the conclusion. This study " The results of this study show that a self-study-based informed decision-making tool increases understanding and agreement with essential elements of HF treatment. In the test group participants, self-study was associated with remaining questions suggesting that a face-to-face encounter with a health-profession should complement the self-study tool."
Major concerns
1. Please explain the novelty of this study.
Response: The novelty of the study is that the COMPARE-EU guided self-study-based information tool shows the appropriateness of this approach while pinpointing that HF patients feel prepared to take a decision after self-study of an information tool but are not ready to take a decision since remaining questions were not answered in the context of the study.
The results suggest that a two-step approach combining unassisted self-study with subsequent face-to-face interaction with a heart failure professional has the potential to increase adherence to heart failure drug treatment.
Furthermore, the results provide the basis for a clinical study comparing adherence outcomes in study participants exposed to the self-study-based information tool with or without subsequent face-to-face encounter with the HF professional.
2. In the study have authors noticed any other metabolic disorders?
Response: Table 2 documents cardiovascular risk factors and interventions. Other comorbidities were not noticed because of the concern that their inclusion into the information tool may weaken the focus on heart failure treatment.
3. The authors must cite more relevant references.
Response: We thank the reviewer and add reference [29]. In addition, we referred to the many publications that accounted for in the meta-analysis by Ruppar et al. [11].
4. In line 76 and 82, start sentences with capital letters.
Response: We thank the reviewer for his attention and changed accordingly.
5. In figure 2 legends, format and styling should be same.
Response: We thank the reviewer for his attention and changed accordingly.
6. In line 225, correct the spacing of usefulness.
Response: We thank the reviewer for his attention and changed accordingly.
7. In table 1, correct the spelling of mixed origin.
Response: We thank the reviewer for his attention and changed accordingly.
Round 2
Reviewer 1 Report
Comments and Suggestions for Authors
THank you for revision. My opinion is that paper is good for publish.
Author Response
Thank you for revision. My opinion is that paper is good for publish.
We thank the reviewer for this kind comment.
Reviewer 3 Report
Comments and Suggestions for Authors
The limitations that I have mentioned in my previous report continue to exist
Author Response
One of the primary weaknesses of this study lies in its limited impact on medication adherence, as the self-study tool improved comprehension but failed to translate into a significant behavioral change. Additionally, the absence of a long-term follow-up restricts the understanding of whether the intervention could sustain adherence or improve clinical outcomes over time.
We thank the reviewer for this comment. The intention of this study was to investigate whether the readiness to adhere to heart failure treatment is improved after exposure to self-study-based information tool without implication of a HF professional. The present study was intended as a pilot-study testing applicability of the self-study-based information tool.
The results of the study confirm applicability of the self-study-based information tool which increased significantly comprehension of the importance of heart failure drug treatment (question 2-4). Furthermore, in the test group, acceptance that "daily intake of HF drugs is consistent with the individual's vision of life" (question 5) was significantly increased when compared the control group. However, readiness to take the decision for adherence failed to improve (question 7) because of remaining questions (question 6). This is an important result suggesting that remaining questions afford answer to achieve rightness of the time for decision-taking.
Questions can knowledge-related or may relate to questions addressing own's intentions and or expectations from clarification of own's short- and long-term goals on the basis of the self-study-based information tool. Therefore, a face-to-face encounter will be valuable complementation offering the opportunity to consider the multimodality of decision-making process to achieve rightness of the time for decision-taking.
Altogether, the pilot study provides the basis for a clinical study comparing adherence outcomes in study participants exposed to the self-study-based information tool with or without subsequent face-to-face encounter with the HF professional.
The small sample size, with only 40 participants in each group, further limits the statistical power and generalizability of the findings.
We agree with the reviewer. However, we would like to point out that this study tested the self-study-based tool with respect to its applicability. The result of the study validates applicability (see answer questions 2-4). In addition, we identified by analysis of different dimensions of the decision-making process (see COMPAR-EU Krah et al.) why the self-study information tool alone failed to empower decision-taking. We agree that this pilot study cannot be taken and copy-pasted into routine follow-up of HF patients. However, the results serve as starting for a future clinical study aiming to provide broadly applicable results.
Furthermore, the study design’s reliance on self-study, without immediate access to healthcare professionals for addressing unresolved questions, may have hindered participants’ confidence in making informed decisions. The slight age difference between the test and control groups (median age: 61 vs. 56 years, p = 0.04) could also act as a confounding variable, potentially affecting outcomes.
We agree with the reviewer for his insightful comment. Indeed, we cannot exclude that the slight age difference may have affected the study results. Because of the small study size, in-depth statistical analysis such as a multivariable analysis does not seem adequate.
Lastly, the narrow focus on HF medication adherence, without considering other critical aspects of disease management such as lifestyle changes or symptom monitoring, restricts the tool's holistic utility in heart failure care.
We thank the reviewer for this comment. In our experience, other critical aspects of HF treatment (salt-intake, restriction of liquid consumption, physical exercise, and others) usually find broader acceptance while need of HF drugs and their potential side effects are routinely put into question resulting in intentional non-adherence in particular in HF patients without sufficient knowledge.